# Can vaccination roll-out be more equitable if population risk is taken into account?

**David R. Sinclair** [1]*, **Asri Maharani**[2], **Daniel Stow**[1], **Claire E. Welsh**[1], **Fiona E. Matthews**[1]

**1** Population Health Sciences Institute, Newcastle University, Biomedical Research Building, Campus for Ageing and Vitality, Newcastle upon Tyne, United Kingdom, **2** Division of Nursing, Midwifery and Social Work, University of Manchester, Manchester, United Kingdom

* David.R.Sinclair@newcastle.ac.uk

## Abstract

### Background

COVID-19 vaccination in many countries, including England, has been prioritised primarily by age. However, people of the same age can have very different health statuses. Frailty is a commonly used metric of health and has been found to be more strongly associated with mortality than age among COVID-19 inpatients.

### Methods

We compared the number of first vaccine doses administered across the 135 NHS Clinical Commissioning Groups (CCGs) of England to both the over 50 population and the estimated frail population in each area. Area-based frailty estimates were generated using the English Longitudinal Survey of Ageing (ELSA), a national survey of older people. We also compared the number of doses to the number of people with other risk factors associated with COVID-19: atrial fibrillation, chronic kidney disease, diabetes, learning disabilities, obesity and smoking status.

### Results

We estimate that after 79 days of the vaccine program, across all Clinical Commissioning Group areas, the number of people who received a first vaccine per frail person ranged from 4.4 (95% CI 4.0-4.8) and 20.1 (95% CI 18.3-21.9). The prevalences of other risk factors were also poorly associated with the prevalence of vaccination across England.

### Conclusions

Vaccination with age-based priority created area-based inequities in the number of doses administered relative to the number of people who are frail or have other risk factors associated with COVID-19. As frailty has previously been found to be more strongly associated with mortality than age for COVID-19 inpatients, an age-based priority system may increase the risk of mortality in some areas during the vaccine roll-out period. Authorities planning COVID-19 vaccination programmes should consider the disadvantages of an age-based priority system.

**Data Availability Statement:** Vaccination distribution data is available from NHS England (https://www.england.nhs.uk/statistics/statistical-work-areas/covid-19-vaccinations). Survey data is

available from ELSA (https://www.elsa-project.ac.uk).

**Funding:** DSi, AM CW and FM are funded by the National Institute for Health Research Policy Research Unit in Older People and Frailty (PR-PRU-1217-21502). DSi, CW and FM are also funded by the Integrated Covid Hub North East. DSt is funded by NIHR School for Primary Care Research (SPCR-PDF-2020-161). The views expressed are those of the author(s) and not necessarily those of of the NIHR, the Department of Health and Social Care, or the Integrated Covid Hub North East.

**Competing interests:** The authors have declared that no competing interests exist.

## Introduction

The COVID-19 pandemic has seen the rapid development of a range of vaccines targeting the SARS-COV-2 virus. Countries are now attempting to obtain and distribute vaccines to their populations in order to reduce the mortality, hospitalisations and social-distancing requirements associated with COVID-19.

Vaccinating entire populations with a limited vaccine supply rate is a time and resource-intensive task. Countries have attempted to optimise the distribution of vaccines by identifying high-risk groups, such as by age and occupation [1–3]. In England, initial vaccines were prioritised for care home residents and staff, followed by a primarily age-based system from the oldest to youngest adults. Clinically extremely vulnerable adults received equal priority with those aged 70-74, while adults in at-risk groups were offered the vaccine only after all adults aged over 65 [4]. Supplies were redirected to maintain parity in these groups between different parts of the country [5]. As high as possible vaccine coverage is important, not only to protect vaccine recipients but also to decrease the likelihood of susceptible individuals encountering infectious people in the general population.

Although age has been shown to be associated with adverse outcomes from COVID-19 [6], age itself is a suboptimal metric of health, with considerable variability in the individual health of people of the same age [7]. These differences are often associated with socio-economic background and health behaviours [8–10].

Frailty is commonly used as a metric of health and vulnerability to adverse outcomes and has been incorporated into primary care records in the English National Health Service (NHS) [9]. There are several different approaches to measuring frailty, however, they all attempt to capture age-related loss of the body's ability to recover from health stressors [7, 10, 11]. Frailty, and the milder classification of 'pre-frailty', have been found to be associated with increased probability of adverse outcomes from COVID-19 [12–16], with frailty more strongly associated with mortality than age [17].

A wide range of risk factors associated with hospitalisation and mortality, and independent of frailty (including atrial fibrillation, chronic kidney disease, diabetes, a learning disability, obesity and smoking status) have also been identified [18], suggesting there may be a large number of people who have greater vulnerability to adverse outcomes from COVID-19, but who may be disadvantaged by vaccine strategies that use age as the primary metric for resource allocation.

In this study we estimate the area-based differences in the vaccination rate of the frail population of England that resulted from an age-based vaccination roll-out. We also evaluate differences based on other risk factors associated with adverse outcomes for risk factors associated with COVID-19. We investigate whether inequities in population health are created by prioritising vaccines primarily by age rather than directly by COVID-19 risk factors associated with adverse outcomes.

## Methods

### Population data

The geographical distribution of the older population was separated into the population seen in each of 135 geographically localised Clinical Commissioning Groups (CCGs). The CCGs reflect the NHS hospital and community based medical services and are responsible for the health of their local population. The age-sex population data [19] and Index of Multiple Deprivation (IMD) rank [20] for each CCG were obtained from the Office of National Statistics. The Index of Multiple Deprivation is England's official measure of relative deprivation for areas. It

synthesises data from income, employment, education, health, crime, housing and the living environment. Lower area ranks are associated with greater deprivation.

## Frailty estimation

The prevalences of frailty and pre-frailty in areas of England was estimated using data from the English Longitudinal Study of Ageing (ELSA) [https://www.elsa-project.ac.uk]. ELSA is a large survey of older people in England which collects information on demographic and socioeconomic characteristics, lifestyle, and health and social care use. ELSA is a prospective cohort study, interviewing every two years. For our analysis, we used ELSA Wave 8 (2016-17), with 8,355 respondents age 50+ and 530 age 85+.

A frailty index was created from each survey using data representing conditions that accumulate with age and are associated with adverse outcomes (S1 File), following established procedures [21–23]. Frailty scores were assigned to each respondent age 50 or over, corresponding to the fraction of all possible frailty 'deficits' reported by each survey respondent. Scores were categorised as robust ($\leq$0.24), pre-frail (>0.24-0.36) or frail (>0.36) using cut-points established in prior literature [9].

An ordinal logistic regression was calculated using each ELSA respondent's age, sex and area deprivation quintile (IMD), with frailty category as the outcome measure. Respondent ages were grouped into 5-year bands from 50-54 to 85-89, with an additional $\geq$90 group. This yielded the probability of a person over 50 being frail, pre-frail or robust, as a function of their age, sex and area deprivation quintile (S2 File). Inverse probability weighting was used to account for sampling and non-response (17.6% [24]). Multiple imputation by chained equations was used to address missing data (S2 File). Potential interactions between frailty, age and sex were investigated and found not to be significant.

Using age and sex-stratified population estimates and IMD rankings for each CCG, we combined the population data with the predicted probability of those individuals being frail to produce estimates of the number of people in each CCG who are expected to be frail. Pre-frailty estimates were similarly calculated.

## Other exposure information

Data on CCG level prevalences of known key risk factors for adverse outcomes associated with COVID-19 infection (atrial fibrillation, chronic kidney disease, diabetes, learning disabilities, obesity, smoking status) [18] were extracted from Public Health England's National General Practice Profiles [https://fingertips.phe.org.uk/profile/general-practice].

## Vaccination data

Information on current levels of vaccination (first dose) in England at the CCG level were taken from NHS England [https://www.england.nhs.uk/statistics/statistical-work-areas/covid-19-vaccinations] on 25 February 2021, the 80th day of the vaccine roll out in England.

## Main analyses

We estimated vaccine take-up by the ratio of the number of people who have received a first dose to 1) the size of the frail population and 2) the size of the older ($\geq$50) population, for each CCG. The higher the number, the more people in the CCG who have received a first vaccine dose relative to 1) the frail and 2) the older populations. Higher ratios for doses per frail person could be caused by a proportionally lower prevalence of frailty, or higher rates of vaccination

administration per CCG. We then present other known risk factors in those CCGs to contrast areas.

## Results

The spread of the vaccine take-up is much less uniform when we consider the additional effect of frailty (Fig 1A) to that of just age (Fig 1B). In particular, vaccination proportions are very low relative to the frail population in parts of the north of England, West Midlands and

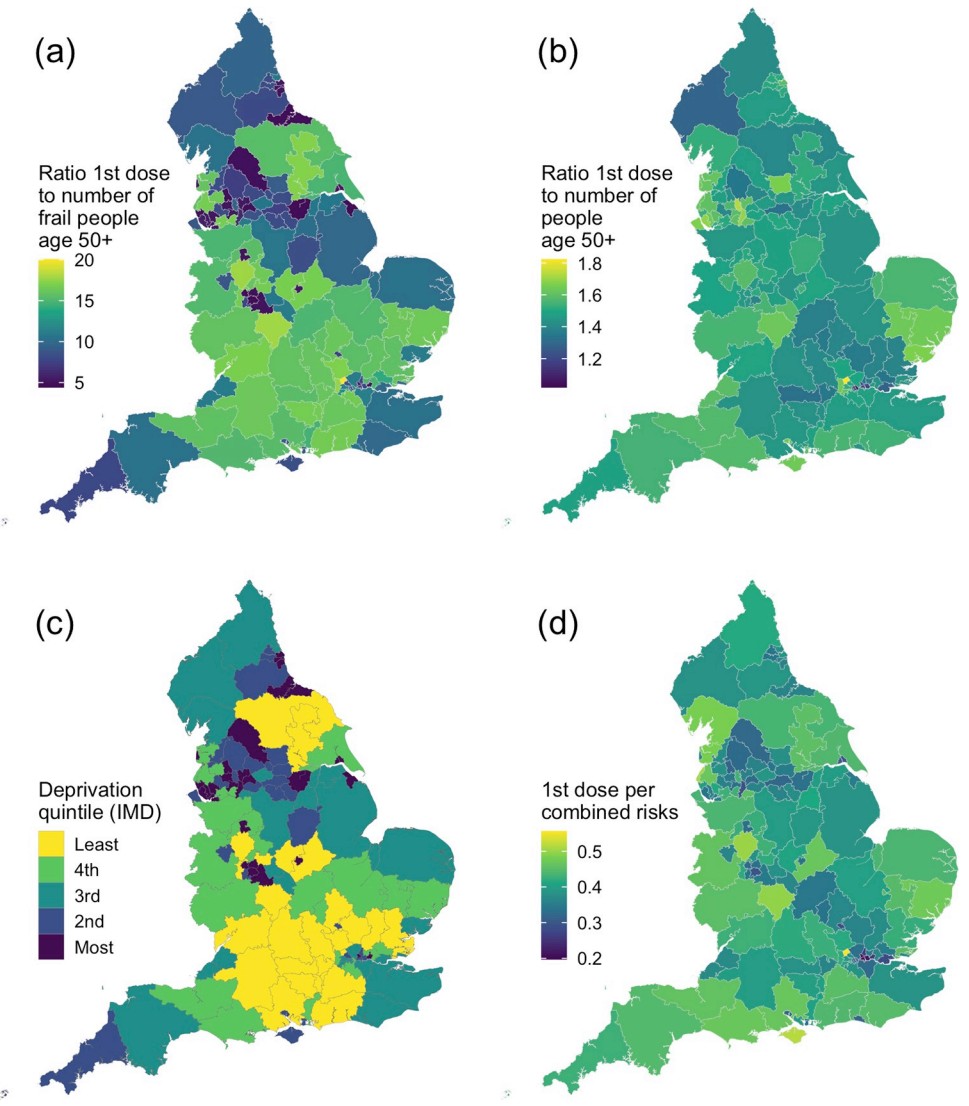

**Fig 1. Vaccination doses relative to the prevalences of COVID-19 risk factors in England.** (a) Ratio of the number of first dose of COVID-19 vaccines administered to the estimated number of frail people over age 50, for each Clinical Commissioning Group area in England. (b) Ratio of the number of first dose of a COVID-19 vaccine administered to the estimated number of people over age 50, for each Clinical Commissioning Group area in England. (c) Area deprivation quintile (measured by English Index of Multiple Deprivation). (d) Ratio of the number of first dose of a COVID-19 vaccine administered to the number of diagnoses for each of seven risk factors associated with adverse COVID-19 outcomes. These risk factors are atrial fibrillation, chronic kidney disease, a learning disability, obesity, smoker and former smoker. They do not include frailty or pre-frailty. Note that this is the total number of diagnoses; a person with multiple risk factors will be counted multiple times for the ratio's denominator.

London. There is a clear negative association between the ratio of first doses to the number of frail people and the area deprivation, as expected (Fig 1C).

The ratio of people who received a first dose of a COVID-19 vaccine to the number of frail people age 50 and over in CCG areas of England varied from 4.4 (95% CI 4.0-4.8) in City and Hackney (inner London) to 20.1 (18.3-21.9) in Harrow (outer London), a factor of 4.6 (4.0-5.2) difference in the ratios. Including both pre-frail and frail people, there is still an approximately 3-fold difference: from 12.7 (9.0-11.9) in Harrow to 4.3 (3.3-4.6) in City and Hackney.

In contrast, when using age as the primary criterion, the ratio of first doses to the number of people over age 50 ranged from 1.8 in Harrow to 1.0 in Central London (Westminster), a much smaller difference in the ratios (Fig 1).

Fig 1D shows how the number of people who received a first vaccine per total diagnoses of a range of risk factors associated with adverse COVID-19 outcomes, varies across all CCGs. The risk factors are atrial fibrillation, chronic kidney disease, a learning disability, obesity, current smoker and former smoker. The number of doses per person with a risk factor ranges from 0.56 (0.54-0.57) in Harrow to 0.20 (0.19-0.20) in Tower Hamlets. S3a-S3g Fig in S3 File maps how the number of first doses administered, relative to the number of people with a range of risk factors, varies across CCGs.

Fig 2 details the top and bottom ten CCGs for first doses per person age over 50 (all CCGs are detailed in S1 Fig). For each of these CCGs, the quintile rank of first doses per person with a range of risk factors for adverse COVID-19 outcomes are also provided. The area deprivation quintile is also displayed. There is an inconsistent relationship between the proportion of first doses administered per person over 50 and the proportion of doses per person with identified risk factors. For example, Berkshire West has one of the lowest numbers of doses per person over 50 (Fig 2A) yet is only in the bottom 40% of areas for two (out of nine) of the investigated risk factors, indicating a lower risk older population than many other areas.

More concerning are areas where low numbers of vaccinations are coincident with high levels of risk (such as Newham and Tower Hamlets). Equally, there are concerning areas in the top CCGs for the number of vaccines (Fig 2B) by age. Areas such as Manchester and Liverpool have some of the highest numbers of vaccines administered per person aged over 50 but are still in the bottom 40% of areas for doses per person at risk for many of the risk factors.

## Discussion

Our analysis suggests that distributing vaccines based only on the age profile of areas does not equitably protect all populations from adverse outcomes associated with COVID-19 infection. England's policy of vaccinating based on age, while easier to manage, is too simple to capture geographic variation in risk. It left some areas with more than four times the number of frail people per vaccination than others. Consequently, these areas may have a greater number of COVID-19 associated hospital admissions and mortality during the vaccine roll-out period. Frailty, as a broad measure of health vulnerability (and is already measured by the NHS), may provide a more equitable means of balancing vaccine distributions.

England's vaccination program commenced on 8 December 2020 [25], with the original aim of offering a first vaccine dose to all adults over 50 or in at-risk groups by 15 April 2021 [26] (although this was achieved 17 March 2021) [27], with a twelve-week delay between the first and second doses. This implies a six-month period in which vaccine protection was inequitably offered.

Additionally, as immune responses to vaccines may be weaker among frail people (as found in studies on non-COVID vaccines [28–30]), their risk of adverse outcomes may be associated with area transmission rates independently of their vaccinations status. Implementing a

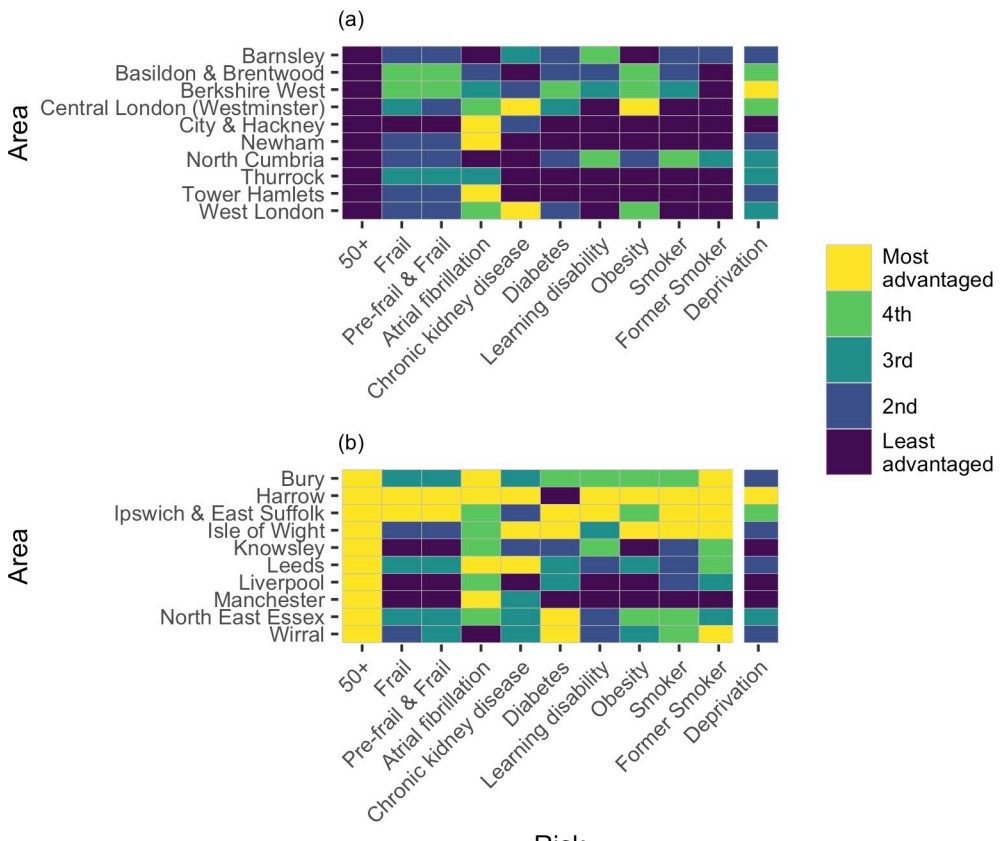

**Fig 2. Heatmap of first vaccine doses administered relative to the number of people diagnosed with risk factors associated with COVID-19, for selected Clinical Commissioning Group (CCG) areas of England.** The (a) bottom and (b) top ten CCG areas for the ratio of first vaccine doses to people aged over 50 are plotted. The ratio of first vaccine dose to the number of people with each of nine risk factors (by quintile), in each area, are shown. The risk factors are associated with infection, hospitalisation and mortality due to COVID-19 [18]. Area deprivation quintile (measured by English Index of Multiple Deprivation) are also shown. Quintiles indicate the most advantaged to least advantaged areas: the most advantaged areas have the most first doses per person with each risk factor and lowest area deprivation.

strategy to increase vaccination rates in disproportionately frail areas, thus decreasing transmission rates in those areas, may help decrease the probability of those who are frail experiencing adverse outcomes. The UK government's reduction of social-distancing requirements uniformly and concurrently across England [26] makes area-based discrepancies in risk especially relevant.

England's COVID-19 vaccination program has been more rapid than most countries, whose programs may be contending with issues such as vaccine accessibility and hesitancy [31]. As vaccine supplies increase, all countries may help optimise their vaccination programs by distributing vaccines using a more focused vulnerability metric than age, such as frailty. Beyond initial COVID-19 vaccinations, it is likely that booster vaccinations and novel vaccines targeting COVID-19 variants will be distributed where possible [32], the distribution of which may be optimised by considerations beyond age.

The age-based approach used in England is simpler to implement than a frailty-based approach, while still correlating with risk factors. Data on the age distributions of areas will be readily available in many countries. An age-based priority is also easier for the population to

understand and thus for individuals to present themselves for vaccination when it is their turn. Additionally, age may be more easily verified at vaccination sites than frailty status.

Although the NHS in England has a frailty measure integrated into its electronic health records [9], this is currently atypical. Additionally, this measure depends on patients having up-to-date primary care records. Most countries would require an analysis, such as the one used here, to estimate the area-based distribution of frailty to conduct a frailty-based roll-out. Area-level frailty prevalence estimates are currently available in several countries [33–35].

Ultimately the aim of a COVID vaccine program is to protect the population, with a focus on the most vulnerable people. If a frailty-based approach can quickly distribute available vaccine supplies and optimises protection for the most at-risk population members, then this is a compelling reason to consider it ahead of an age-based system.

Using area-based frailty metrics would be simpler to manage than rather than individual frailty measures. Although it would only indirectly increase the vaccination rate among the most vulnerable, at-risk population members would benefit from greater immunity in the surrounding population, decreasing their risk of infection (possibly more so than vaccination for those with poor antibody responses). If vaccines were prioritised on an individual level using a frailty metric, it may be more difficult to rapidly vaccinate large numbers of frail people due to potentially poorer health, limited mobility and temporary medical ineligibility for vaccination due to illnesses or other medical treatment.

Our analysis found a positive association between increased frailty prevalence and increased area deprivation. This positive association was imposed when using area demographics to estimate the number of frail subjects (S2 File). This implies that targeting vaccines preferentially in deprived areas may also be beneficial for protecting people who are frail. This may have the advantage of being already measured in places where frailty is not. However, as the relationship between frailty prevalence and area deprivation may vary between countries, this may make a vaccine roll-out based on deprivation less effective. Additionally, targeting vaccines at the most deprived may have the unintended consequence of stimulating vaccine hesitancy among a part of the population which may already have increased vaccine hesitancy [36].

This analysis has limitations. Frailty prevalence is based on synthetic estimates, which account for demography and socio-economic data, but not localised health behaviours. Data availability meant we were only able to consider the over 50 frail population. The COVID risk factors are a selection of those identified [18] which have CCG-level data available from Public Health England; it does not consider other risk factors, such as ethnicity and vaccine hesitancy [18, 37]. An ideal metric may use individual-level data that includes all risk factors.

## Conclusions

While the distribution of COVID-19 vaccines in England has managed to approximately balance the number of doses administered per person over the age of 50 across the country, variability in doses per frail person is much wider: a factor of over three between some areas. This variability is also apparent for other COVID-19 risk factors. Balancing vaccine distribution by the prevalence of frailty across areas may be more equitable in protecting the most vulnerable in future vaccine rollouts.

## Supporting information

**S1 File. Variables included in the frailty index.**
(PDF)

**S2 File. Statistical analysis.**
(PDF)

**S3 File. Number of first vaccine doses administered, relative to the number of people diagnosed with a range of risk factors, in each Clinical Commissioning Group area.**
(PDF)

**S1 Fig. Heatmap of first vaccine doses administered relative to the number of people diagnosed with risk factors associated with COVID-19, for all Clinical Commissioning Group (CCG) areas of England.** The risk factors are associated with infection, hospitalisation and mortality due to COVID-19. The ratio of first vaccine dose to the number of people with each risk factor (by quintile), in each area, are shown. Area deprivation quintiles (measured by the English Index of Multiple Deprivation) are also shown. All Clinical Commissioning Group areas in England are plotted.
(PDF)

## Author Contributions

**Conceptualization:** David R. Sinclair, Daniel Stow, Fiona E. Matthews.

**Data curation:** David R. Sinclair, Asri Maharani.

**Formal analysis:** David R. Sinclair, Asri Maharani.

**Funding acquisition:** Fiona E. Matthews.

**Investigation:** David R. Sinclair.

**Methodology:** David R. Sinclair, Asri Maharani, Fiona E. Matthews.

**Supervision:** Fiona E. Matthews.

**Visualization:** David R. Sinclair.

**Writing – original draft:** David R. Sinclair.

**Writing – review & editing:** David R. Sinclair, Asri Maharani, Daniel Stow, Claire E. Welsh, Fiona E. Matthews.

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
