## [Decision Letter · Decision Letter 0]

10 Aug 2021

PONE-D-21-13238

Can vaccination roll-out be more equitable if population risk is taken into account?

PLOS ONE

Dear Dr. Sinclair,

Thank you for submitting your manuscript to PLOS ONE. After careful consideration, we feel that it has merit but does not fully meet PLOS ONE’s publication criteria as it currently stands. Therefore, we invite you to submit a revised version of the manuscript that addresses the points raised during the review process.

ACADEMIC EDITOR:

Please take into account Reviewer suggestions, to make the complex statistical background more easlily accessible to the readers, including the ones less familiar with UK NHS. In particular I would advice Authors to give more information regarding the calculation of Index of Multiple Deprivation and provide data regarding IMD quintiles in different CCGs, possibly adding a column in Figure 2, as suggested by Reviewer 2, and in Figure A4. (By the way, if I have understood well, the term “IMD quantile” used in Appendix should probably be replaced with “IMD quintile”). Moreover It should be better clarified how IMD was used at the individual level in ELSA and CFAS sample, to build the frailty prediction model.

Moreover, as suggested by Reviewer 1, English policy regarding vaccination should better clarified (moreover ref 23 is no longer accessible on the internet). If priority groups changed after 25-FEB-2021 (and this seems to be the case according to https://www.gov.uk/government/publications/covid-19-vaccination-care-home-and-healthcare-settings-posters/covid-19-vaccination-first-phase-priority-groups) this should be briefly discussed, as it seems to be consistent with Authors suggestions.

As further suggestions, I feel that the aim (lines 87-89) should be more precisely stated and that the 10-year time interval between ELSA and CFAS data and the pandemic should be cited as a study limitation in the estimate of frailty.

We look forward to receiving your revised manuscript.

Kind regards,

Enrico Mossello

Academic Editor

PLOS ONE

1. Please ensure that your manuscript meets PLOS ONE's style requirements, including those for file naming. The PLOS ONE style templates can be found at https://journals.plos.org/plosone/s/file?id=wjVg/PLOSOne_formatting_sample_main_body.pdf and https://journals.plos.org/plosone/s/file?id=ba62/PLOSOne_formatting_sample_title_authors_affiliations.pdf.

2. Please expand the acronym “NIHR” (as indicated in your financial disclosure) so that it states the name of your funders in full.

3. Acknowledgments Section: Move New Information to the Financial Disclosure:

"Thank you for stating the following in the Acknowledgments Section of your manuscript:

“DSi, FM, CW and AM who are funded by the National Institute for Health Research Policy Research Unit in Older People and Frailty (PR-PRU-1217-21502). DSi, CW and FM are also funded by the Integrated Covid Hub North East. DSt is funded by NIHR School for Primary Care Research (SPCR-PDF-2020-161). The views expressed are those of the author(s) and not necessarily those of the funders: Integrated COVID Hub North East (ICHNE), NIHR or the Department of Health and Social Care. CFAS II was funded by UK Medical Research Council (MRC; research grant G0601022) and the Alzheimer’s Society.”

“DSi, FM, CW and AM who are funded by the National Institute for Health Research Policy Research Unit in Older People and Frailty (PR-PRU-1217-21502; https://www.nihr.ac.uk/). DSi, CW and FM are also funded by the Integrated Covid Hub North East. DSt is funded by NIHR School for Primary Care Research (SPCR-PDF-2020-161; https://www.nihr.ac.uk/). The funders had no role in study design, data collection and analysis, decision to publish, or preparation of the manuscript.”

Additional Editor Comments (if provided):

Reviewers' comments:

Reviewer's Responses to Questions

**Comments to the Author**

1. Is the manuscript technically sound, and do the data support the conclusions?

Reviewer #1: Yes

Reviewer #2: Yes

2. Has the statistical analysis been performed appropriately and rigorously? 

Reviewer #1: Yes

Reviewer #2: Yes

3. Have the authors made all data underlying the findings in their manuscript fully available?

Reviewer #1: No

Reviewer #2: Yes

4. Is the manuscript presented in an intelligible fashion and written in standard English?

Reviewer #1: Yes

Reviewer #2: Yes

5. Review Comments to the Author

Reviewer #1: This is a well written manuscript w/ a great message. I would suggest the following in addition to strengthen the paper:

1. Please describe in more detail in the intro the current policy in the UK for vaccine distribution. In the conclusions the authors note that they use age, but this isn't clear. More detail is needed.

2. Please add a heatmap of vaccines : patients with at least one risk factor to fig 1 as third panel

3. If you have data on COVID-19 infection rates (pre-vaccination) and/or mortality by county and could show if vaccine distribution/uptake maps to areas of higher cases and/or mortality that would also be of interest

Reviewer #2: I appreciated the opportunity to review this paper which examines whether vaccination rollout could be more equitable if based on a better measure of population risk, rather than using primarily age. The authors combined data on the number of first vaccine doses administered across 135 NHS Clinical Commissioning Groups in England with area-based estimates of the proportions of the population aged 65+ and living with frailty. The frailty estimates were generated using a frailty index approach in the ELSA and CFAS II national surveys. Heat maps are presented contrasting vaccine distribution by both area age and area frailty, and also by a selection of other known COVID risk factors. The authors identify inequalities in the number of vaccine doses administered relative to the number of people who are frail or have other risk factors and emphasize that these inequalities are exacerbated when using a simple age-based approach. They therefore recommend that authorities planning COVID-19 vaccination programs should consider the disadvantages of using an age-based priority system.

Overall this paper is thought-provoking and makes an interesting contribution to the literature. It is well written and uses novel, creative and appropriate methods to examine these important questions. I have the following comments:

1. I am very sympathetic to the argument that a more targeted approach based on frailty makes good sense. The age-based approach does have some practical advantages, in that age provides some approximation of risk and a much easier and non-arbitrary criteria to use in a large (and in many ways unprecedented) vaccination program rollout. The authors could expand on the practicality of using a frailty-based approach. Using the frailty index in England could be more feasible than in most other jurisdictions given the integration of the frailty measure in the Electronic Health Record which is (unfortunately) not a feature of most EMRs and health systems. Of course, it would be nice to aspire to bring other jurisdictions up to better including frailty measures in the health record rather than aiming for the lowest common denominator. Overall, it would be helpful for the authors to acknowledge both the pros and the cons of the age-based approach in more detail, with particular emphasis on practicalities.

2. In a related question, how would the use of a frailty-based approach work? Would it be done on an individual level where a person is invited to book an appointment based on their known frailty level or shows their frailty value in the vaccine clinic booking process in order to gain access to the vaccination? What about people who have not had access to health care to assess their level of frailty – would they run the risk of being passed over? Or would it be done at an area level where the population % with frailty would be considered and vaccines would be preferentially delivered to clinics in areas of high frailty to be then distributed in whatever way is most practical? This would also potentially have the benefit of creating a herd protection in areas of high vulnerability while keeping the rollout relatively less complex. In either case, it would be helpful to clarify which approach is being suggested.

3. The figures are helpful and interesting but I imagine they would be most easily interpreted by people with prior knowledge of social comparisons between these areas of England. Might the authors consider including socioeconomic status in the figures for easy reference? A third panel c could be included for the map showing the area variation in the Index of Multiple Deprivation for example in Figure 1. In Figure 2, another column could be added to each of the heat maps showing the IMD. The authors could consider whether this would help them enrich the context of their figures.

4. This brings up the point that presumably the inequity in vaccine access arises at least in part because of socioeconomic status and including SES more prominently in the paper and figures would help make that clear. However, one wonders if this could be offset because in general lower socioeconomic status areas may have a higher density of frontline workers who may also have been prioritized in the early phases of vaccination. Indeed, it’s likely that a solid argument could be made that targeting vaccine uptake in lower socioeconomic areas would have a similar benefit to targeting areas with high frailty. The authors may consider strengthening their justification of why targeting by frailty is more important than targeting by SES in this way.

5. The use of the terms (in)equity and (in)equality could be clarified since they have different technical meanings.

6. PLOS authors have the option to publish the peer review history of their article (what does this mean?). If published, this will include your full peer review and any attached files.

Reviewer #1: No

Reviewer #2: **Yes: **Melissa K Andrew

---

## [Author Response · Author response to Decision Letter 0]

20 Sep 2021

Comments from Reviewer #1

1. Please describe in more detail in the intro the current policy in the UK for vaccine distribution. In the conclusions the authors note that they use age, but this isn't clear. More detail is needed.

Thank you for highlighting this. We have added more details on vaccine policy in England in the Introduction section (lines 63-66 in the version of the manuscript without tracked changes).

2. Please add a heatmap of vaccines: patients with at least one risk factor to fig 1 as third panel.

This has been added as Fig 1d. 

3. If you have data on COVID-19 infection rates (pre-vaccination) and/or mortality by county and could show if vaccine distribution/uptake maps to areas of higher cases and/or mortality that would also be of interest

We agree that this is an interesting issue worth investigating, however the reasons behind vaccine hesitancy are beyond the scope of this paper and might be better considered in a separate study.

Comments from Reviewer #2

1. I am very sympathetic to the argument that a more targeted approach based on frailty makes good sense. The age-based approach does have some practical advantages, in that age provides some approximation of risk and a much easier and non-arbitrary criteria to use in a large (and in many ways unprecedented) vaccination program rollout. The authors could expand on the practicality of using a frailty-based approach. Using the frailty index in England could be more feasible than in most other jurisdictions given the integration of the frailty measure in the Electronic Health Record which is (unfortunately) not a feature of most EMRs and health systems. Of course, it would be nice to aspire to bring other jurisdictions up to better including frailty measures in the health record rather than aiming for the lowest common denominator. Overall, it would be helpful for the authors to acknowledge both the pros and the cons of the age-based approach in more detail, with particular emphasis on practicalities.

We thank you for this suggestion. As a results, we have expanded the Discussion to include these points (lines 227-242 in the version of the manuscript without tracked changes).

2. In a related question, how would the use of a frailty-based approach work? Would it be done on an individual level where a person is invited to book an appointment based on their known frailty level or shows their frailty value in the vaccine clinic booking process in order to gain access to the vaccination? What about people who have not had access to health care to assess their level of frailty – would they run the risk of being passed over? Or would it be done at an area level where the population % with frailty would be considered and vaccines would be preferentially delivered to clinics in areas of high frailty to be then distributed in whatever way is most practical? This would also potentially have the benefit of creating a herd protection in areas of high vulnerability while keeping the rollout relatively less complex. In either case, it would be helpful to clarify which approach is being suggested.

We have added to the discussion to consider this question (lines 244-251).

3. The figures are helpful and interesting but I imagine they would be most easily interpreted by people with prior knowledge of social comparisons between these areas of England. Might the authors consider including socioeconomic status in the figures for easy reference? A third panel c could be included for the map showing the area variation in the Index of Multiple Deprivation for example in Figure 1. In Figure 2, another column could be added to each of the heat maps showing the IMD. The authors could consider whether this would help them enrich the context of their figures.

We have added an area deprivation map to Figure 1 (Fig 1c). We have also added the deprivation quintile to Figure 2.

4. This brings up the point that presumably the inequity in vaccine access arises at least in part because of socioeconomic status and including SES more prominently in the paper and figures would help make that clear. However, one wonders if this could be offset because in general lower socioeconomic status areas may have a higher density of frontline workers who may also have been prioritized in the early phases of vaccination. Indeed, it’s likely that a solid argument could be made that targeting vaccine uptake in lower socioeconomic areas would have a similar benefit to targeting areas with high frailty. The authors may consider strengthening their justification of why targeting by frailty is more important than targeting by SES in this way.

This is an interesting point. We have included a paragraph to the Discussion that considers targeting vaccines by SES (lines 253-260).

5. The use of the terms (in)equity and (in)equality could be clarified since they have different technical meanings.

Thank you for highlighting this discrepancy. We have clarified this.

---

## [Editor Report · Decision Letter 1]

25 Oct 2021

PONE-D-21-13238R1Can vaccination roll-out be more equitable if population risk is taken into account?

PLOS ONE

Dear Dr. Sinclair,

Thank you for submitting your manuscript to PLOS ONE. After careful consideration, we feel that it has merit but does not fully meet PLOS ONE’s publication criteria as it currently stands. Therefore, we invite you to submit a revised version of the manuscript that addresses the points raised during the review process.

ACADEMIC EDITOR: The Authors have modified the analysis and provided full information, in agreement with Reviewers' request. I feel that some minor changes would still be useful:

- lines 159-161: add "as expected" to the following statement ("There is a clear negative association between the ratio of first doses to the number of frail people and the area deprivation")

- accordingly, modify discussion (lines 253-254), specifying that the present analysis model ineherently implies a positive association between increased frailty prevalence and increased area deprivation, as area deprivation was used to estimate the number of frail subjects

- clarify statistical model (lines 124-125), stating that frailty category was used as outcome measure.

We look forward to receiving your revised manuscript.

Kind regards,

Enrico Mossello

Academic Editor

PLOS ONE
---

## [Author Response · Author response to Decision Letter 1]

28 Oct 2021

Academic Editor comments

1. lines 159-161: add "as expected" to the following statement ("There is a clear negative association between the ratio of first doses to the number of frail people and the area deprivation").

Thank you for this suggestion. We have added “as expected” to the above sentence, such that it now reads: “There is a clear negative association between the ratio of first doses to the number of frail people and the area deprivation, as expected (Fig 1c)" (lines 160-162).

2. accordingly, modify discussion (lines 253-254), specifying that the present analysis model inherently implies a positive association between increased frailty prevalence and increased area deprivation, as area deprivation was used to estimate the number of frail subjects.

We agree that it is helpful to make the relationship between area deprivation and frailty clear in the discussion. We are concerned that “specifying that the present analysis model inherently implies a positive association between increased frailty prevalence and increased area deprivation” may be misleading to some readers. The ordinal logistic regression fitted to the survey data does not necessarily imply a positive association between frailty and deprivation, rather this association is a result of the regression. This positive association is then imposed when estimating the number of frail subjects in each area.

Accordingly, we have amended this section (now lines 254-256) to read: “Our analysis found a positive association between increased frailty prevalence and increased area deprivation. This positive association was imposed when using area demographics to estimate the number of frail subjects (Appendix 2).”

3. clarify statistical model (lines 124-125), stating that frailty category was used as outcome measure.

Thank you for suggesting this improvement to the clarity of the manuscript. We have added that frailty category is the outcome measure of the statistical model in lines 124-125. The relevant sentence now reads: “An ordinal logistic regression was calculated using each ELSA respondent’s age, sex and area deprivation quintile (IMD), with frailty category as the outcome measure”.

---

## [Editor Report · Decision Letter 2]

2 Nov 2021

Can vaccination roll-out be more equitable if population risk is taken into account?

PONE-D-21-13238R2

Dear Dr. Sinclair,

We’re pleased to inform you that your manuscript has been judged scientifically suitable for publication and will be formally accepted for publication once it meets all outstanding technical requirements.

Kind regards,

Enrico Mossello

Academic Editor

PLOS ONE
---

## [Editor Report · Acceptance letter]

5 Nov 2021

PONE-D-21-13238R2 

Can vaccination roll-out be more equitable if population risk is taken into account? 

Dear Dr. Sinclair:

I'm pleased to inform you that your manuscript has been deemed suitable for publication in PLOS ONE. Congratulations! Your manuscript is now with our production department. 

Kind regards, 

on behalf of

Dr. Enrico Mossello 

Academic Editor

PLOS ONE